



**Dynamic Complex Network Analysis of PM$_{2.5}$ Concentrations in the UK**
**using Hierarchical Directed Graphs (V1.0.0)**
Parya Broomandi [1,2,6]
Xueyu Geng [1]
Weisi Guo [3,1,4]
Jong Ryeol Kim [2]
Alessio Pagani [4]
David Topping [5,4]
[1] School of Engineering, The University of Warwick, Coventry, CV4 7AL, UK.
[2] Department of Civil and Environmental Engineering, Nazarbayev University, 010000,
Astana, Kazakhstan.
[3] School of Aerospace, Transport, and Manufacturing, Cranfield University, Bedford, MK43
0AL, UK.
[4] The Alan Turing Institute, London, UK.
[5] School of Earth, Atmospheric and Environmental Science, University of Manchester,
Manchester M13 9PL, UK.
[6] Department of Chemical Engineering, Masjed-Soleiman Branch, Islamic Azad University,
Masjed-Soleiman, Iran.







*Abstract*

Worldwide exposure to fine atmospheric particles can exasperate the risk of a wide range of
heart and respiratory diseases, due to their ability to penetrate deep into the lungs and blood
streams. Epidemiological studies in Europe and elsewhere have established the evidence base
pointing to the important role of $PM_{2.5}$ (fine particles with a diameter of 2.5 microns or less) in
causing over 4 million deaths per year. Traditional approaches to model atmospheric
transportation of particles suffer from high dimensionality from both transport and chemical
reaction processes, making multi-sale causal inference challenging. We apply alternative
model reduction methods – a data-driven directed graph representation to infer spatial
embeddedness and causal directionality. Using $PM_{2.5}$ concentrations in 14 UK cities over a 12-
month period, we construct an undirected correlation and a directed Granger causality network.
We show for both reduced-order cases, the UK is divided into two a northern and southern
connected city communities, with greater spatial embedding in spring and summer. We go on
to infer stability to disturbances via the network trophic coherence parameter, whereby we
found that winter had the greatest vulnerability. As a result of our novel graph-based reduced
modeling, we are able to represent high-dimensional knowledge into a causal inference and
stability framework.

Key words: complex network; atmospheric pollution; $PM_{2.5}$



## 1. Introduction:

1.1 Background and rationale

Atmospheric particulate matter can be attributed to both local emissions (by both stationary and mobile sources) and regional transport processes. Causal inference between primary (emitted directly by the emission sources) and secondary (produced in the atmosphere by the transformation of gaseous pollutants) is challenging. For example, whilst combustion sources such as road traffic account for the bulk of anthropogenic PM emissions and cause $PM_{2.5}$ formation (Munir, 2017; AQEG, 2012), meteorological conditions can also influence $PM_{2.5}$ concentrations through dispersion, and deposition. Due to the high data complexity and dimensionality caused by the contribution of atmospheric chemistry transport processes and a range of emission sources in ambient $PM_{2.5}$ concentrations, we need to overcome the high dimensionality challenge and compress the concentration data into 2-dimensional (2D) network. European legislation sets current and future caps on anthropogenic emissions of primary and secondary-precursor components of $PM_{2.5}$ at national level and from individual sources (Vieno et al., 2016). In addition, it is well-known that ambient PM derives from both transboundary emissions and transport (Vieno et al., 2016), creating challenges to develop effective mitigation scenarios at the local level (Vieno et al., 2016; Zhang et al., 2008; van Donkelaar et al., 2010).

1.2 Importance & Impact

Atmospheric particulate matters impact human health (WHO, 2006, 2013) and climate change through radiative forcing (IPCC, 2013). The global health burden from exposure to ground level $PM_{2.5}$ is substantial. According to the Global Burden of Disease project, exposure to ambient $PM_{2.5}$ concentrations prevailing in 2005 was responsible for 3.2 million premature deaths and 76 million disability-adjusted life years (Vieno et al., 2016; Lim et al., 2012). In Europe, exposure to ambient $PM_{2.5}$ is still a major health issue. For the period 2010–2012, it was reported by the European Environment Agency report that 10–14 % of the urban population in the EU28 countries were exposed to $PM_{2.5}$ exceeding the EU annual-mean $PM_{2.5}$ reference value ($25 \, \mu g \, m^{-3}$), while 91–93 % were exposed to concentrations exceeding the WHO annual-mean $PM_{2.5}$ ($10 \, \mu g \, m^{-3}$) (Gehrig et al., 2003; EEA, 2014). Meeting the standards focused on $PM_{2.5}$ is complicated by the considerable chemical heterogeneity. PM long-term exposure has been identified to be more serious than the daily (short-term) exposure to higher PM concentrations that was first linked to impacts on human health (Pope and Dockery, 2006; Harrison et al., 2012). Long-term impact studies have provided the foundation for calculation





of health impacts from PM exposure in the UK and Europe, which are significant
(COMEAP,2010). Changes in the direction of studies towards $PM_{2.5}$, associated with the
evidence that long-term PM levels play important role alongside short-term peaks, in terms of
health outcomes, has caused changes in legislation (Defra, 2007, Official Journal, 2008).

1.3 Modeling Challenges
Challenges associated with traditional modelling of PM evolution to infer regional and local
influences include the need to embed a chemical complexity, range of emission sources and
transformative processes in Eularian models. In this study, for the first time, we explore the
potential for compressing ambient $PM_{2.5}$ network data into 2-dimensional (2D) network,
establishing a simple graph to infer causality and stability.  This is a timely study as strategic
investments in national and local air quality monitoring networks require an evaluation on the
usefulness, or not, of network design. Whilst this study focuses on a sparse distributed network,
we discuss future applications for local networks across cities, for example. In a graph, each
node in the graph is a city, which exhibits a temporal signal ($PM_{2.5}$) and is connected to other
cities if they exhibit a close association in terms of either correlation (undirected) or Granger
causality (directed).

**2. Materials and Methods:**
2.1 Ground-level $PM_{2.5}$ data




Hourly $PM_{2.5}$ concentrations were observed at 15 monitoring stations in different cities (from
UK-air defra dataset website[1]) shown in Figure 1 and coordinates given in SI – List S1. The
study period was divided into four seasons (meteorological seasons) Spring: 1st March 2017-
31st May 2017, Summer: 1st June 2017- 31st August 2017, Autumn: 1st September 2017- 30th
November 2017, and Winter: 1st December 2017- 28th February 2018. Also, $PM_{2.5}$ emissions
sources data were downloaded from the UK National Atmospheric Emission Inventory (NAEI)
website.

UK monitoring Stations

Figure 1. Studied stations in the UK.

2.2 Cross correlation calculation for spatial distribution of $PM_{2.5}$ in the UK
To measure the similarity of $PM_{2.5}$ concentration time series among each pair of cities in the
current study, the hourly based cross-correlation (XCROSS) was calculated using PAST
(PAleontological Statistics) version 3.25, for all site pairs (106 pair of cities) in four seasonal
windows (spring, summer, autumn, and winter). These periods were selected to try and capture
the effect of seasonal changes on the measured similarity between $PM_{2.5}$ concentration time

---

[1] https://uk-air.defra.gov.uk/data/openair

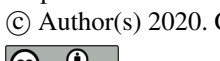



series. A flexible threshold (above 70%) was applied to decide which pairs were strongly
correlated (Gehrig et al., 2003).
2.3 Granger Causality calculation in PM$_{2.5}$ network in the UK
The Granger causality test as a statistical hypothesis test for determining whether one time
series is useful in forecasting another, thus for measuring the ability to predict the future values
of a time series using prior values of another time series, was applied (using Eviews, version
11) to each pair of cities in the network during different seasons. When the p-value was less
than alpha level (5%), the null hypothesis was rejected, and we could decide which time series
can forecast another one. The Granger Causality test assumes that both the $x$ and y time series
(x and y represent PM$_{2.5}$ concentration series for different stations in our network) are
stationary, which was not the case in current study. As a result, de-trending was first employed
before using the Granger Causality test. To retain the same degrees of freedom (Statistical
parameter estimation is based on different amounts of data or information. The number of
independent pieces of data that go into the estimation of a parameter are called the degrees of
freedom (DF). Mathematically, DF represents the number of dimensions of the domain of
a random vector, or how many components should be known before the vector is fully
determined.), with annual data, the lag number is typically small (1 or 2 lags). For quarterly
data (which was our case), the appropriate lag number is 1 to 8. If monthly data is available, 6,
12, or 24 lags can be used given enough data points. The number of lags is critical since a
different number of lags can lead to different test results. As a result, optimal lags were chosen
based on Akaike Information Criterion (AIC). The optimal lag number that ensures the model
will be stable is thus 7 in our study. It is possible that causation is only in one direction, or in
both directions ($x$ Granger-causes y and y Granger causes $x$). We chose the direction based on
the lowest p-value. For example in spring, according to our analysis,  results suggest that
'activity' in Manchester is statistically influencing Preston with a p-value= $5{\times}10^{-29}$, while
Preston is statistically affecting Manchester with a p-value= $3{\times}10^{-8}$. Therefore we infer that the
first statement (pollution from Manchester is influencing Preston's concentrations) is the
correct one to select due to its lower p-value.  Please note the language chosen reflects the
statistical inference for the network analysis; However, the mapping of inference to
atmospheric behavior and known challenges around PM$_{2.5}$ source apportionment is important
and discussed.

2.4 Trophic coherence



Trophic coherence is a way of hierarchically restructuring a directed network and labelling the
hierarchical levels (trophic levels – as derived from food webs and predation levels). Trophic
levels have been shown to be an effective compressed metric to infer stability on large directed
networks with no clear input output definition. The bottom (basal) nodes are those where all
energy comes from (e.g. major source of pollution), and the coherence of the whole network is
a proxy for stability against disturbances. The trophic level ($s_i$) of a node i, is the mean trophic
level of its in-neighbours:

$$s_i = 1 + \frac{1}{k_i^{in}} \sum_j a_{ij} s_j$$

where $k_i^{in} = \sum_j a_{ij}$ is the number of in-neighbours of the node i and $a_{ij}$ is the adjacency matrix
of the graph. Basal nodes $k_i^{in}$ have trophic level $s_i = 1$ by convention (Pagani et al., 2019). In
our study, to interpret trophic coherence in a directed causal network, the initial stage was
introducing basal nodes.

Stations with a low trophic level are PM$_{2.5}$ sources while stations with a high trophic level are
receptors according to this definition. The trophic level of a station is the average level of all
the stations from which it receives PM$_{2.5}$ pollutant plus 1. $x_{ij} = s_i - s_j$ is the associated trophic
difference of each edge. As always, p(x) (the distribution of trophic differences) has a mean
value of 1, and when the network is more trophically coherent, the variance of this distribution
is smaller. The incoherence parameter q is the measurement of the trophic coherence of
network, which is the standard deviation of p(x):

$$q = \sqrt{\frac{1}{L} \sum_{ij} a_{ij} x_{ij}^2 - 1} \,,$$

where $L = \sum_{ij} a_{ij}$ is the edges (the number of connections) between the nodes (stations) in the
network. When $q = 0$, the network is perfectly coherent however q with the values of greater
than 0 shows less coherent networks.

**3. Result and Discussion:**

3.1 Spatial distribution of PM2.5 over the UK

Interesting information about the spatial distribution of the PM$_{2.5}$ concentrations over the UK
can be obtained when analysing the cross correlation of the hourly values between the different





sites. Results suggest that two groups of cities were connected to each other with XCROSS
value above 70%. The first group (Northern Group A) includes Preston (Pre), Manchester
(Man), Chesterfield (Chest), Leeds, Nottingham (Not), Newcastle (New), Birmingham (Bir),
and Liverpool (Liv), while the second one (Southern Group B) includes Bristol (Bri), Oxford
(Oxf), Southampton (South), Plymouth (Ply), Norwich (Nor), and London (two stations named
LonB and LonR). For the seasons of spring, summer, and autumn, the combination of groups
does not change, but the value of XCROSS does (Figure 2). In wintertime the combination of
cities in and out of clusters changes (Figure 2-D). The connected cities, generating a directed
dynamic network, are seasonally visualized in Figure 2.
As the networks are very spatial (i.e., distance is a significant impedance factor), a general
measure of how spatially embedded it is, was studied. The pair of stations were divided into
groups based on the distance (Table 1). To quantify the level of spatial embeddedness, a
relationship between Cross correlation and distance between each pair of cities was studied
(Table 1). A very high spatially embedded part of the network for all seasons was formed below
100 Km, while less spatial embeddedness of network was witnessed when the distance
increased to above 200Km (for all seasons). A main part of the network (100 Km) was formed
in cluster A with percentage of 67%, 54%, 60%, and 89% during spring, summer, autumn, and
winter, respectively. This value in cluster A reduced (for all seasons) when increasing the
distance between pair of cities reaching the value of zero during autumn and winter. Since the
distance between cities in cluster was dominantly above 100Km, the dominant part of the
network in cluster B was formed below 200 Km (100-200Km), with percentage of 38%, 52%,
46%, and 23% during spring, summer, autumn, and winter, respectively. This value in cluster
B had a reduction (for all seasons) by increasing the distance between pair of cities reaching
the value of zero during autumn, while during wintertime it was 19% for distance above
200Km. The number of outliers (pair of connected cities out of group A &B) had its highest
values of 40%, 100%, and 81% during spring, autumn, and winter, respectively when the
distance between cities was above 200Km. During autumn, for distances above 200Km, the
original network was not formed, while during winter, group B was formed. The number of
paired cities in the network had a reduction by 50% between spring and winter, when the
distance was below 100Km (the same reducing trend was witnessed in both groups). For
distances below 200Km, the network was weakened by %50. Interestingly, when the distance
between cities increased above 200 Km, during winter the network was strengthened by 17%
comparing to spring.



Table 1. The relationship between Cross-Correlation (XCROSS) of the daily values of $PM_{2.5}$ and distance of the cities in UK.

| Distance | Pair of connected cities in network | Pair of connected cities in group A | Pair of connected cities in group B | Outliers (pair of connected cities out of groups) |
|---|---|---|---|---|
| **Spring** | | | | |
| <100Km | 18 (43%) | 12 (67%) | 6 (33%) | 0 |
| <200Km | 42 (81%) | 24 (57%) | 16 (38%) | 2 (5%) |
| >200Km | 10 (19%) | 3 (30%) | 3 (3%) | 4 (40%) |
| **Summer** | | | | |
| <100Km | 13 (52%) | 7 (54%) | 6 (46%) | 0 |
| <200Km | 25 (90%) | 12 (48%) | 13 (52%) | 0 |
| >200Km | 3 (10%) | 2 (67%) | 1 (33%) | 0 |
| **Autumn** | | | | |
| <100Km | 15 (54%) | 9 (60%) | 6 (40%) | 0 |
| <200Km | 28 (93%) | 9 (27%) | 13 (46%) | 9 (27%) |
| >200Km | 2 (7%) | 0 | 0 | 2 (100%) |
| **Winter** | | | | |
| <100Km | 9 (35%) | 8 (89%) | 1 (11%) | 0 |
| <200Km | 26 (41%) | 14 (54%) | 6 (23%) | 6 (23%) |
| >200Km | 37 (59%) | 0 | 7 (19%) | 30 (81%) |

**A**

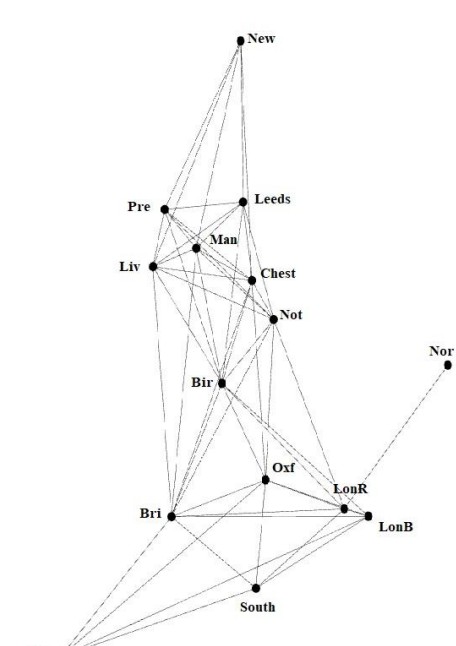





































**D**

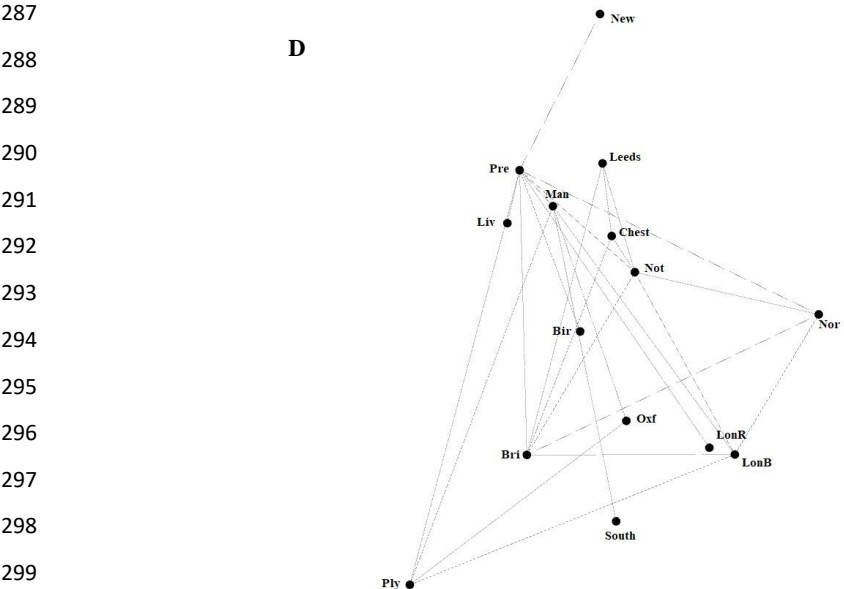

Figure 2. Cross correlation based dynamic network including; A) spring window, B) summer window,
C) autumn window, and D) winter window in 2017-2018, UK.



3.2 Granger causality test
The main result from this study is that cities with the strongest Cross correlation have the lowest
p-value (below 5%) (Figure 3). In spring, as already noted, results suggest that, statistically,
activity in Manchester is causing concentrations to change in Preston with p-value= $5\times10^{-29}$
(i.e. Manchester $PM_{2.5}$ data can be used to predict the future $PM_{2.5}$ values of Preston) and
Bristol is causing Oxford with a p-value of $9\times10^{-28}$. In summer, Liverpool is causing Preston
with a p-value of $7\times10^{-17}$. Manchester is causing Preston with p-value= $6\times10^{-23}$ in autumn,
while Chesterfield is causing Nottingham with a p-value of $1\times10^{-7}$in wintertime. The results
look very spatial and the distance is a significant impedance factor. The distance between all
paired cities was below 50Km. Based on Table 2, when the distance between pair of cities
increases the order of p-value increases too.





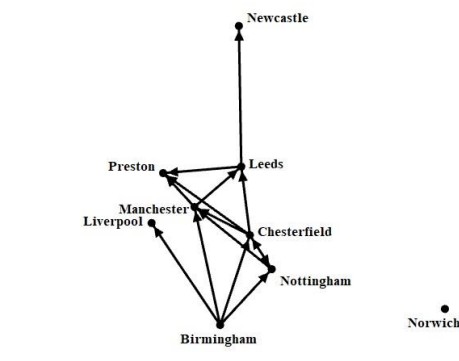



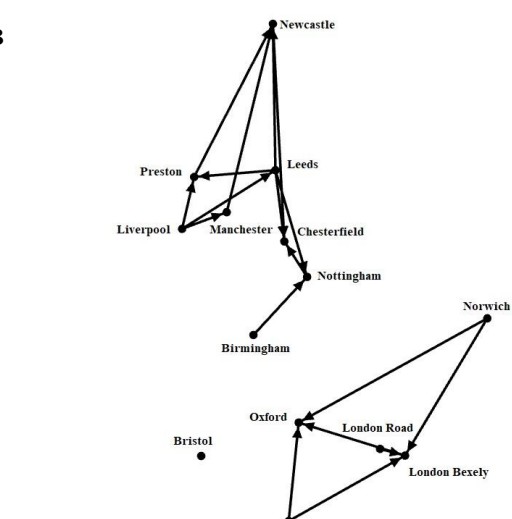






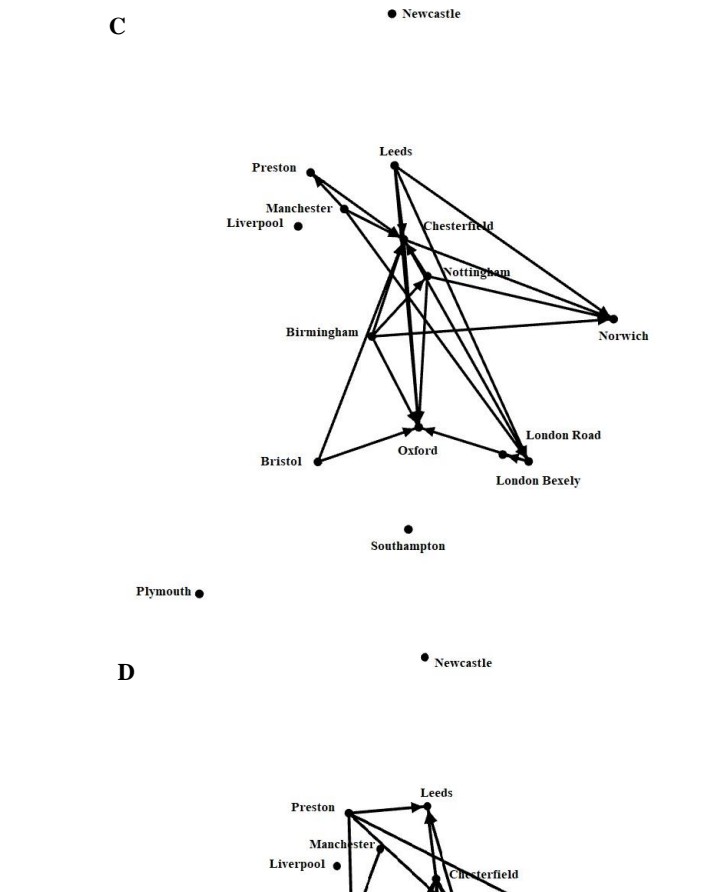



Figure 3. Granger based dynamic network including; A) Spring window, B) Summer window, C)
Autumn window, and D) Winter window in 2017-2018, UK.




Table 2. Comparison among Granger causality results (p-values) in different seasons.

| Source | Target | Distance (Km) | p-value |
|---|---|---|---|
| **Spring** | | | |
| Manchester | Preston | 43.66 | $5\times10^{-29}$ |
| Bristol | Oxford | 91.78 | $9\times10^{-28}$ |
| **Summer** | | | |
| Liverpool | Preston | 42.62 | $7\times10^{-17}$ |
| Leeds | Newcastle | 131 | $5\times10^{-11}$ |
| **Autumn** | | | |
| Manchester | Preston | 43.66 | $6\times10^{-23}$ |
| Chesterfield | Oxford | 165.11 | $3\times10^{-20}$ |
| **Winter** | | | |
| Chesterfield | Nottingham | 36.17 | $1\times10^{-7}$ |
| Chesterfield | Bristol | 213.74 | $7\times10^{-6}$ |


A directed graph is defined (Bang-Jensen and Gutin, 2008) as an ordered pair $G = (N, E)$,
where N is a set of nodes (i.e. stations) and E is a set of ordered pairs of nodes, called edges
(i.e the probability values for F statistics). The hierarchical structure of a directed graph can be
presented by its trophic coherence property. The whole idea is that hierarchical systems have
fewer feedback loops and experience less cascade effects. The incoherence parameter (q) was
used to measure the coherence of the seasonal causal network to show how trophic distance is
tightly associated with edges concentrated around its mean value (which is always 1) (Johnson,
et al., 2014). We observed incoherent network in our seasonal datasets (Table 3).

Table 3. Incoherence factor of seasonal directed networks in current study.

| Directed network | Incoherence factor (q) |
|---|---|
| Spring | 0.69 |
| Summer | 0.37 |
| Autumn | 0.49 |
| Winter | 0.35 |


The highly incoherent season was spring with q= 0.69, whilst a less incoherent network was
found to be winter (q=0.35). In figure 3, according to the parameter definition, the basal nodes
with the low trophic level represent the major pollution source nodes, while stations with high
trophic levels are ones who act as receptors in the causal network. During springtime, due to
well mixing of the lower atmospheric layer, the network was well formed. In group A,



Birmingham with low trophic level was classified as a pollution source, while in group B
Southampton was pollution source with low trophic level.

**A**

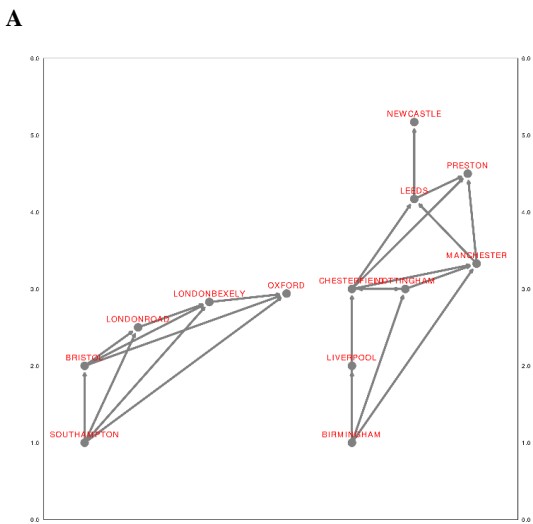


**B**

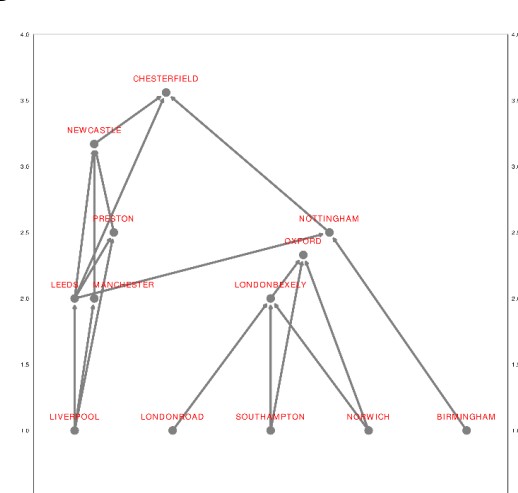











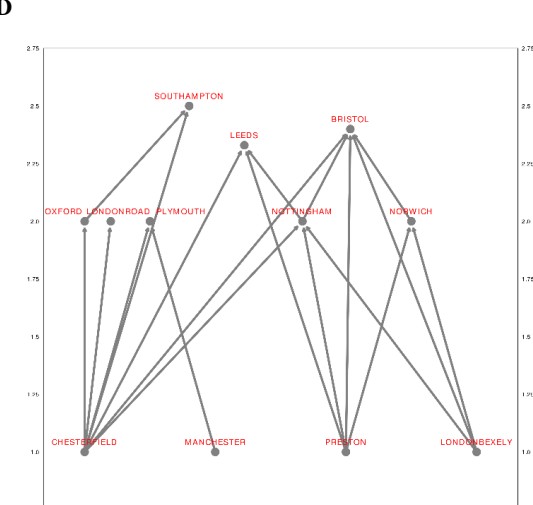



Figure 3. The hierarchical structure and basal nodes of causal network including; A) Spring window, B) Summer window, C) Autumn window, and D) Winter window in 2017-2018, UK.







## 4. Discussion:

*4.1 The effect of meteorological parameters on network structure*

Based on the previous analysis, this connection (network) indicates that meteorological conditions and diurnal emissions from a wide range of common sources (such as traffic), rather than locally specific sources and events, dominate the relative variations of the concentrations of fine particles over long periods (Gehrig et al., 2003). During wintertime, the meteorology is characterized by frequent inversions, forming an efficient obstacle for the distribution and homogenization of PM. As a result, only tight spatially embedded parts of network (below 100Km with the highest percentage of restored network) could 'withstand' meteorological influences and further parts (above 100Km) started to collapse from a network perspective. In winter time, the plausible reason of connecting the cities out of the initial network (81% of connected cities were out of the initial network with distance above 200 Km) might be higher average seasonal wind speeds (in all studied stations), probably due to the balance among greater dilution and shorter transport times at higher wind speeds, which allows less time for PM dispersion and deposition over further distances (Harrison et al., 2012).

Indeed, it is well known that changes in meteorological parameters (e.g., wind speed and direction, temperature, and rainfall) can significantly affect $PM_{2.5}$ concentrations and formation mechanisms (AQEG, 2012; Vieno, et al., 2016). In addition to primar$_y$ sources, secondary sources are dependent on meteorological conditions and the abundance of precursors. Secondary aerosols have a significant contribution in $PM_{2.5}$ concentrations in the UK, where a large proportion transboundary secondary $PM_{2.5}$ transferred from Europe is made of nitrate particles in the form of ammonium nitrate (AQEG, 2012; Vieno, et al., 2016). One plausible reason of connection within a network can be common transboundary sources.

The association among wind direction and $PM_{2.5}$ can provide a better picture of the origins of the measured $PM_{2.5}$ concentrations. With this in mind, there is an outstanding coherence throughout the patterns across the Group A and Group B in the UK. Hence, there is, a minor variation between cities in the south (Group B) and those in the north or close to northern part (Group A) of the UK (Harrison et al., 2012). High $PM_{2.5}$ concentrations in Group B (southern sites) are more attributed to winds from the east through to southeast, which are often attributed to a blocking high pressure over the Nordic countries, giving rise to a south-easterly or easterly air flow that cause transportation of emissions from eastern Europe, northern Germany, and the Belgium and Netherlands to the southern cities in the UK (Harrison et al., 2012; Barry and



Chorley, 2010). Nonetheless, the arriving air flow in the northern parts of the UK from the east
to southeast sector will not have passed through these same emission origins.
On the other hand, High $PM_{2.5}$ concentrations in Group A (northern cities or close to northern
part) are more important attributed to the winds blowing from the northeast through to east,
drawing air flow ( likely to start blowing when a low pressure runs up the English Channel)
northward across European emission sources (to mainly be emission sources of precursors of
secondary PM), out into the North Sea, then reaching northern parts of the UK from a north-
easterly direction (Barry and Chorley, 2010).

**5.  Conclusion:**

In current study, we use $PM_{2.5}$ concentrations in 14 cities in the UK over 52 weeks to infer an
undirected correlation and a directed Granger causality network. We show for both network
cases (group A & B), two robust spatial communities divide the UK into the northern and
southern city clusters, with greater spatial embedding in spring and summer.
Based on the granger causality test, we infer that $PM_{2.5}$ data of cities with the strongest Cross
correlation (having the lowest p-value) can be helpful to predict the future $PM_{2.5}$ values in the
network. However, there are of course multiple caveats with this statement, some of which are
reflected in our discussions around known influences from meteorological and source
variability. We leverage on the directed network to infer stability to disturbances via the trophic
coherence parameter, whereby we found that winter had the greatest vulnerability.
As already noted, this connection (network) suggests that meteorological conditions and
emissions from regional origins rather than specific local origins and events dominate the
relative variations of the urban background $PM_{2.5}$ concentrations (Gehrig et al., 2003) using
this sparse network data. We know that PM with emission sources from continental Europe,
probably as secondary PM, can play an important role in affecting $PM_{2.5}$ levels in different
parts of the UK (Harrison et al., 2012). However, our study has some limitations including a
short period of time over which the network was analysed. Also, to have a better understanding
of network, evaluating a predictive network based $PM_{2.5}$ model using meteorological
parameters, and contributions from identified clusters in the UK, would be helpful.  This work
acts as a demonstrator for the information that can be extracted from an undirected correlation
and a directed Granger causality network. Further work is needed, alongside ancillary data that
might support the extracted relationships such as source apportionment data and transport



activity, for example. The approach might also be better suited to more local networks, such as
monitoring stations across a city.
*Code availability.* The code for computing the trophic level of each node in the network, the
trophic difference and finally trophic coherence (q) of the network with all scripts needed to
reproduce the results in this study is available at https://github.com/kohyar88/PM2.5--Trophic-
-Coherence-/tree/v1.0.0 with DOI number of 10.5281/zenedo.3661483.
**Acknowledgement:**
This project has received funding from the European Union's Horizon 2020 research and
innovation programme Marie Skłodowska-Curie Actions Research and Innovation Staff
Exchange (RISE) under grant agreement No. 778360.

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
