# Peer review of "Dynamic Complex Network Analysis of PM$_{2.5}$ Concentrations in the UK using Hierarchical Directed Graphs (V1.0.0)"

_Geoscientific Model Development, 2019_

## Referee Comment (RC1) · Anonymous Referee #1 · 16 Apr 2020

Broomandi et al. present a graph theoretical analysis of causality and PM2.5. They demonstrate the utility of network analysis for understanding complex behavior of atmospheric pollutants. The work applies a novel technique in the analysis of PM2.5 data across the United Kingdom, with results consistent with domain knowledge. While the results are potentially scientifically interesting, the presentation of the manuscript is difficult to read, and it is ultimately challenging to assess the quality of this work. This is particularly the case as it relates to the assessment of the impact of meteorology on PM2.5 using these causal networks. I cannot recommend this manuscript for publication in GMD at this time.

More specific comments are below:

**References and Introductory Motivation**

It is not clear from the manuscript what problem the authors are intending to solve with this work. They mention the need to "overcome the high dimensionality challenge" on line 70, but there are plenty of existing tools for studying the high dimensionality in atmospheric composition, chiefly among them chemical transport models. What does this work provide that more traditional simulation experiments cannot?

Is it instead a demonstration of a new technique in atmospheric pollution research? If so, the authors should state this and demonstrate its utility in relationship to existing knowledge.

The references are quite sparse for this manuscript. Additional background and motivational clarity should include more details of previous applications of causality in the geosciences (e.g. Ebert-Uphoff, Imme, and Yi Deng. "Causal Discovery for Climate Research Using Graphical Models." Journal of Climate).

**Materials and Methods**
**Section 2.1**

The description of the data in this section was inadequate for assessing the quality of the research in this work. How often were measurements taken? How was averaging done? What instruments were used? Was data quality assessed in any way? Is there a DOI or citation appropriate for any of the data used?

**Section 2.2**
The software used (e.g. PAST, EVIEW) should be appropriately cited.

Line 139: What is this threshold for, how is it selected, and how is it calculated?

**Section 2.3**
This section outlines a set of methods unfamiliar to the majority of the geoscientific modelling community. I suggest the authors include more details and relevant citations for the broader community should they be inclined to dig deeper into this sort of analysis. At the very least, this section should be carefully edited for clarity. The large number of parenthetical elements

throughout the section make it challenging to parse what is actually being stated. The three sentences within a parenthetical statement on lines 150-155 are emblematic of this.

Line 149: Detrending a dataset does not always correct for non-stationarity. Please cite a reference for its appropriateness here.

Line 157: What is a lag used for in this case?

Lines 345-348: It isn't clear in the text why this basic description of a directed graph is in the middle of the section on Granger Causality?

**Section 2.4**
The language regarding trophic coherence and sources/sinks is unreferenced and thus assumed to be an innovation of this work. All cities in this region are known to be atmospheric sources of PM2.5 due to anthropogenic and natural processes. The discussion here and elsewhere in the manuscript of only some cities classified as sources of PM2.5 is at odds with reality and should be clarified further.

While tropic coherence has been shown to provide interesting results for the analysis of food webs, it's not clear from the text that this is the most appropriate method for assessing important vertices in a causal graph for PM2.5. Why was it used here?

On line 39 in the abstract, the authors claim that winter is the most coherent of the seasons. They attribute this to meteorological features like wind speeds and inversions. Table 2 shows that summer has nearly the same incoherence factor as winter, and yet this is not discussed at all in the manuscript. Given the vastly different meteorological features in summertime, does this influence the interpretation that meteorology is a driving factor?

**Discussion**
In general, this discussion seems incomplete and largely conjecture without appropriate referencing or analysis presented herein.

Lines 378-381: What in the previous analysis indicates that meteorological conditions and diurnal emissions from regional sources dominate? I acknowledge that this is known the be case throughout the field a priori, but it's not clear how this analysis in this manuscript leads to that conclusion.

Lines 399-409: How are these relationships known without detailed trajectory modeling? The fact that the causal networks are consistent with known transport mechanisms is interesting but does not provide evidence for these sweeping assessments of pollutant transport.

Lines 410-415: There is not evidence presented in this manuscript that this transport mechanism is attributable to PM2.5 variability in the Northern UK.

**Conclusions**
Line 423-425: The authors haven't shown any results related to predicting future PM2.5 between cities.

Line 439-431: This work is not the first to demonstrate that meteorology drives much of the variability in PM2.5. It is not clear from this manuscript how connections beyond conjecture can be made to meteorological variability and PM2.5.

**Minor Comments**
Line 78: "Atmospheric particulate matters" should be "Atmospheric particulate matter"

Figure captions should contain more detail regarding the content of the figured. For example in Figure 2, what do edge thicknesses correspond to?

Line 232: "%50" should be "50%".

Line 393: "Primar$_y$" should be "Primary".

---

## Referee Comment (RC2) · Anonymous Referee #2 · 24 Apr 2020

The authors have used two statistical techniques to look at how in particulate matter (PM) concentration in different cites over the UK are linked. The methods they use are an undirected correlation method and a Granger causality network. The core point they argue to explore is the groupings of cities within the data whether a time-series in one could predict another. The data they use is a 1-year subset of freely available UK PM data for 15 sites. This has been considered with emissions data from the UK National Atmospheric Emission Inventory (NAEI).

I have concerns about the possible artefacts in the analysis and even the scientific validity of the results due to: (1) the small subset of data used (e.g. 15 sites in 14 cities),

(2) the small temporal period considered (just 1 year), (3) small about of parameters considered (solely PM observations and a national inventory of emissions), (4) the lack of consideration of known variables that influence PM (e.g. meteorology), (5) the insufficient of placing this work in the context of existing related publications

Importantly, I am not convinced that authors have shown the efficacy of this method and how it could be used for the policy goals they motivate the study with. I have written some specific points below. However, this study would need to be greatly expanded, all points addressed, an general presentation/readability improved to be suitable for publication in GMD. I cannot recommend publication of this manuscript.

Many years of data are freely available for hundreds of sites across the UK. Why have only 15 sites from 14 locations been considered here? Why were these 15 sites (listed in the SI) chosen above the hundreds of others? Are the authors saying they are representative of the sites? Why only 15 sites? Is it due to limitations in a computer resource or does this approach not scale well? If this is the case could a representative average be taken of the sites in a city? Why has a mixture of types been used (e.g. Traffic urban, Background suburban, Background Urban, and Industrial Urban)? What impact could the choice of data set have on the conclusions? All of these issues need to be explored through sensitivity testing. This analysis must be expanded in breadth (# sites) and depth (e.g. years of data) if it is to be considered a case study or proof of concept is.

How many air quality events happened in this year? What does this approach show about these events? Often pollution events in the UK are synoptic in scale, so would affect multiple cities. It would be interesting to see what information could be drawn out from this analysis.

The discussion needs to be expanded to explain what the pros and cons of this approach are over the generally taken approaches (e.g. chemical transport models). Also consider this approach against other non-explicit approaches taken in this field

(e.g. Nowack et al 2020, Keller et al 2019). ML methods that consider much more input data (e.g. meteorology, emissions, chemistry) have already been demonstrated in a much more thorough way within this journal (e.g. Keller et al 2019). Also make it clear what new information that this approach will provide and why this would be advantageous above these other approaches or complementary to them.

Line 102 - This is sentence is far too strong. The authors have not demonstrated that this technique could be this useful.

"This is a timely study as strategic investments in national and local air quality monitoring networks require an evaluation on the usefulness, or not, of network design"

Line 112 - See earlier point about the limitations of data used.

Line 117 - Was this emissions inventory used for the same year? Given more details here.

Line 163 - What is the link to the broader picture here? If we accept that a background urban site in Manchester can be used to predict future concentrations in Preston, is the suggestion that something like this could be done for other sites and a national level of prediction gained at a computational cheap cost?

Line 424 - There is no supporting evidence given for this predictive capacity

Technical points

Title / other text - Some lines are highlighted blue. Why?

Table 2 - typo in units? (Kelvin metres instead of kilometres?)

Figure 1 - resolution needs to be improved. Most of UK coastline should be shown for context. Which map layer was used?

Figure 2 - resolution needs to be improved.

Figure 3 - resolution needs to be substantially improved. Axis labels are not large

enough or readable.

Line 9 - Specify the region over which the deaths occur. Europe? Global?

Line 32 - Have people not already been successful in doing this? (e.g. the climate science community - Nowack et al 2020)

Line 34 - "12 month", "52 week" are used interchangeably in the manuscript. . . would a single phrasing (e.g. year-long) be easier for the reader?

Line 35 - Sentence does not scan. superfluous "two" and "a"?

Line 80 - Globally? The focus of the article is the UK, so you need to be more specific.

Line 112 - This time-split would be better placed in a table and in the SI. It is not a very readable way of presenting this information.

Line 116 - Was data downloaded for the same period. This sentence does not need to start with "also".

Line 393 - formating error of "y" in word primary.

Data availability - Add a data availability section. Include specific details on which version of data was used and how to access this.

References - remove full stops at starts of lines.

S1 list - please update the information to be in a table format. The current format is unwieldy, not read-friendly.

Citations

Keller, C. A. and Evans, M. J.: Application of random forest regression to the calculation of gas-phase chemistry within the GEOS-Chem chemistry model v10, Geosci. Model Dev., 12, 1209–1225, https://doi.org/10.5194/gmd-12-1209-2019, 2019.

Nowack, P., Runge, J., Eyring, V. and Haigh, J.D., 2020. Causal networks for climate

model evaluation and constrained projections. Nature communications, 11(1), pp.1-11.

---

## Author Comment (AC1) · 29 May 2020

29/05/2020 Dear GMD editorial board, Subject: Submission of revised paper gmd-2019-342 Thank you for your email dated 1st May 2020 enclosing the reviewers' comments. We have carefully reviewed the comments and have revised the manuscript accordingly. Our responses are given in a point-by-point manner below. Changes to the manuscript are highlighted. We hope the revised version is now suitable for publication and look forward to hearing from you in due course. Sincerely, Jong R. Kim, PhD, PE, CMP Professor Department of Civil and Environmental Engineering School of Engineering and Digital Sciences Nazarbayev University Nursultan (Astana), Kaza-

khstan

Phone: +7 7172 70 9136 Mobile: +7 775 069 1735

gmd-2019-342-RC1#

• They mention the need to "overcome the high dimensionality challenge" on line 70, but there are plenty of existing tools for studying the high dimensionality in atmospheric composition, chiefly among them chemical transport models. What does this work provide that more traditional simulation experiments cannot? Is it instead a demonstration of a new technique in atmospheric pollution research? If so, the authors should state this and demonstrate its utility in relationship to existing knowledge.

Response: (Updated in Paper to the discussion section): We are happy to provide more context here. For sure we are demonstrating a new technique that may be used alongside more established/traditional methods. The general framing of our approach is at the national level, trying to demonstrate (via a data-driven correlation and causal network), the statistical relationship between pollution data between multiple cities. This data-driven low-dimensional network enables us to examine seasonal trends and infer root causal mechanisms. This, we believe, is of a much lower complexity than deploying a chemical transport model of UK, where inference of causality remains challenging. Of course, what our model lacks is the relationship back to the physical flow models, and our future work would like to connect the models together.

We also respond in more detail (not in paper): As the reviewers are well aware, chemical transport models require emission inventory data (local or regionally originated) and a meteorological core to predict the dispersion and deposition of pollutants such as PM2.5. Beside the notable amount of required data, high performance computing [HPC] platforms are required to deploy and evaluate model outputs, not least including experience with the pre and post processing software environments. In the current study we attempt to investigate the behaviour of PM2.5, using a 2-dimensional (2D) network constructed from observational data alone. This leads to, first, the correlation

network, and then a causation network. We can identify two things of note. (1) is the presence of root causations of pollution in certain seasons across a large region of UK, and (2) the stability of the transport network to potential disturbances. Both provide a level of simplistic insight at a very low complexity.

  The references are quite sparse for this manuscript. Additional background and motivational clarity should include more details of previous applications of causality in the geosciences (e.g. Ebert-Uphoff, Imme, and Yi Deng. "Causal Discovery for Climate Research Using Graphical Models." Journal of Climate).

Response: The proposed study and also some other related references will be addressed in revised version of our manuscript. (Updated in the Paper to the discussion section): To infer causality, correlation-based methods such as lagged linear regression are already used in climate variability studies. This method can provide valuable information about causal relationships, but is susceptible to overreporting significant relationships when one or more of the variables has substantial autocorrelation (memory)(Ebert-Uphoff and Deng, 2012; McGraw and Barnes, 2018; Runge et al., 2017). On the other hand, Granger causality considers the autocorrelation of data and as a result is not susceptible to overreporting significant relationships. Since Granger causality is straightforward to calculate, it can be a preferred option to traditional lagged regression analyses when one or more datasets has substantial autocorrelation (memory). In addition, the establishment of a relationship between two variables is not sufficient in determining the true causality, but also determining the direction of causality is also needed; A more difficult task and challenge to overcome. The correlation-based methods cannot provide any information regarding directionality (but are still popular and useful for identifying lagged relationships among climate variables). However, the Granger approach has its own limitations as it does not account for mediating variables or indirect effects. Also, it requires assumptions of stationary and linear processes(Davidson et al., 2016; McGraw and Barnes, 2018; Wang et al., 2015, 2004). In previous studies, Nowack et al. (2020) showed that causal model evaluation provides

stronger relationships for constraining precipitation projections under climate change as compared to traditional evaluation metrics for precipitation or storm tracks(Nowack et al., 2020). As a result, casual network analyses could be a promising tool to constrain long-term uncertainties in climate change projections. When a method relies on the assumption that previous model skill can be related to projected future changes will definitely suffer from certain limitations, including; the existence of some processes which are not ( or not well) represented in current climate models and might become important in the future, and there is possibility that not all of relevant processes be well captured through the studied model(Nowack et al., 2020).

Materials and Methods Section 2.1 The description of the data in this section was inadequate for assessing the quality of the research in this work. How often were measurements taken? How was averaging done? What instruments were used? Was data quality assessed in any way? Is there a DOI or citation appropriate for any of the data used?

Response:  How often were measurements taken? The measurement are taken from UK Automatic Urban and Rural Network (AURN) (https://uk-air.defra.gov.uk/data/openair). More information about UK Automatic Urban and Rural Network is available online from the DEFRA website(DEFRA, 2015).

 How was averaging done? Data coverage were checked to have minimum missing data and having at least 75% of hourly based measured data for all stations, before averaging the hourly PM2.5 concentration. Only available data for 20 hours a day were averaged. While zero, NAN, and negative values were removed from the data set, and if the remained values were at least 20 hours a day, we averaged it representing the daily PM2.5 concentration, if not we report that day as NAN.

 What instruments were used?

For PM2.5 measurement in UK monitoring system, for daily and hourly averaged concentrations, the instrument of FAI SWAM 5a was used by Defra (Defra approved

instrument) which was certified to MCERTS (The Environment Agency's Monitoring Certification Scheme) for UK particulate Matter, and also certified to MCERTS for CAMs (Continuous Ambient Measurement Systems) of particulate Matter (https://uk-air.defra.gov.uk/networks/monitoring-methods?view=mcerts-scheme) (DEFRA, 2015). Reference equivalent method FDMS (Filter Dynamic Measurement System) is used for PM2.5 measuring at studied stations, which is allowed by EU for regulatory purposes(AQEG, 2012).

 Was data quality assessed in any way?

Characterisation of PM2.5 temporal variability is important when it can help us to observe the high levels of pollutant causing health problems. Due to the data unavailability in the UK, it is not possible to conduct the historically long-term temporal trend analysis of PM2.5 (AQEG, 2005). Based on the AQEG (2012) report, there are no monitoring stations with long term (> 5years) using reference equivalent instruments for PM2.5 monitoring. From 2008–2009 onward, with the increase in the number of monitoring stations using reference equivalent method (such as FDMS allowed by EU for regulatory purposes) it is possible to study the temporal changes in PM (PM10 & PM2.5) in the UK (Munir, 2016). Minimum performance requirement for PM10 & PM2.5 analysers were outlined in standard method of EN12341:2014 PM10 and PM2.5 (EN16450:2017 Automatic PM analysers). These methods are proposed to ensure that measurement methods are complying with the DQO (Data Quality Objectives) set down in the Ambient Air Quality Directive (2008/50/EC) and in the amending Directive (EU) 2015/1480. The monitoring techniques used the UK's AURN for PM10 & PM2.5 ( with the exception of the automatic PM10 analysers) are; Tapered Element Oscillating Microbalance, Beta Attenuation monitor, Gravimetric monitor, Filter Dynamics Measurement System (FDMS), Optical light scattering, and Fine dust Analysis System (FIDAS)(DEFRA, 2015).  Is there a DOI or citation appropriate for any of the data used?

Yes, there is: Department for Environment, Food and Rural Affairs, United Kingdom:

http://uk-air.defra.gov.uk/, Last Access: 27 July 2015.

(Updated in the paper to the data availability section): Data availability: The measurements are taken from UK Automatic Urban and Rural Network (AURN) (https://uk-air.defra.gov.uk/data/openair). More information about UK Automatic Urban and Rural Network is available online from the DEFRA website (DEFRA, 2015). Data coverage were checked to have minimum missing data and having at least 75% of hourly based measured data for all stations, before averaging the hourly PM2.5 concentration. Only available data for 20 hours a day were averaged. While zero, NAN, and negative values were removed from the data set, and if the remained values were at least 20 hours a day, we averaged it representing the daily PM2.5 concentration, if not we report that day as NAN. For PM2.5 measurement in UK monitoring system, for daily and hourly averaged concentrations, the instrument of FAI SWAM 5a was used be Defra (Defra approved instrument) which was certified to MCERTS (The Environment Agency's Monitoring Certification Scheme) for UK particulate Matter, and also certified to MCERTS for CAMs (Continuous Ambient Measurement Systems) of particulate Matter (https://uk-air.defra.gov.uk/networks/monitoring-methods?view=mcerts-scheme) (DEFRA, 2015). Reference equivalent method FDMS (Filter Dynamic Measurement System) is used for PM2.5 measuring at studied stations, which is allowed by EU for regulatory purposes(AQEG, 2012). Characterization of PM2.5 temporal variability is important when it can help us to observe the high levels of pollutant causing health problems. Due to the data unavailability in the UK, it is not possible to conduct the historically long-term temporal trend analysis of PM2.5 (AQEG, 2005). Based on the AQEG (2012) report, there are no monitoring stations with long term (> 5years) using reference equivalent instruments for PM2.5 monitoring. From 2008–2009 onward, with the increase in the number of monitoring stations using reference equivalent method (such as FDMS allowed by EU for regulatory purposes) it is possible to study the temporal changes in PM (PM10 & PM2.5) in the UK(Munir, 2016). Minimum performance requirement for PM10 & PM2.5 analysers were outlined in standard method of EN12341:2014 PM10 and PM2.5 (EN16450:2017 Automatic PM analyzers). These methods are proposed to

ensure that measurement methods are complying with the DQO (Data Quality Objectives) set down in the Ambient Air Quality Directive (2008/50/EC) and in the amending Directive (EU) 2015/1480. The monitoring techniques used the UK's AURN for PM10 & PM2.5 ( with the exception of the automatic PM10 analyzers) are; Tapered Element Oscillating Microbalance, Beta Attenuation monitor, Gravimetric monitor, Filter Dynamics Measurement System (FDMS), Optical light scattering, and Fine dust Analysis System (FIDAS)(DEFRA, 2015).

Section 2.2 • The software used (e.g. PAST, EVIEW) should be appropriately cited.

Response:  PAST software: (Hammer et al., 2001)

 Eviews (version 11) software: (Software, 2019)

• Line 139: What is this threshold for, how is it selected, and how is it calculated?

(Updated in the Paper to the Materials & Methods section): Based on previous similar study conducted in Switzerland to characterize the spatial distribution and seasonal changes of PM2.5 and PM10 concentrations using long-term monitoring data (Gehrig and Buchmann, 2003), we decided to choose 70% as our threshold cross-correlation.

Section 2.3 This section outlines a set of methods unfamiliar to the majority of the geoscientific modelling community. I suggest the authors include more details and relevant citations for the broader community should they be inclined to dig deeper into this sort of analysis. At the very least, this section should be carefully edited for clarity. The large number of parenthetical elements throughout the section make it challenging to parse what is being stated. The three sentences within a parenthetical statement on lines 150-155 are emblematic of this.

Response: The section is revised based on valuable comment provided by reviewer. (Updated in the paper to the Materials & Methods section): The Granger causality test statistically ascertains if one time series can cause the other. Thus to see that prior values of a time series contain the information about the future values of another time series. This method was applied (using Eviews, version 11)(Software, 2019) to each pair of cities in the network during different seasons. Following this, statistically significant results (p<0.05) were used to determine which time series contain information about the future values of another. The Granger Causality test assumes that both x and y time series (x and y represent PM2.5 concentration series for different stations in our network) are stationary, which was not the case in current study. As a result, de-trending was firstly employed before using the Granger Causality test(Papagiannopoulou et al., 2017, 2016). To retain the same degrees of freedom (DF) (mathematically, DF represents the number of dimensions of the domain of a random vector, or how many components should be known before the vector is fully determined.), with annual data, the lag number is typically small (1 or 2 lags). For quarterly data (in our case), the appropriate lag number is 1 to 8. If monthly data is available, 6, 12, or 24 lags will be used given enough data points. The number of lags is critical since a different number of lags lead to different test results. Consequently, the optimal lag number of 7 ensures the stability of model in this case study (based on Akaike Information Criterion (AIC). There is possibility of causation in one or both directions (x Granger-causes y and y Granger causes x). The chosen direction was based on the lowest p-value. For example according to our analysis, in spring we infer that 'activities' in Manchester is statistically influencing concentrations measured in Preston with a p-value= $5 \times 10$-29, while Preston is statistically affecting Manchester with a p-value= $3 \times 10$-8. Therefore, the first statement (pollution from Manchester is influencing Preston's concentrations) is the correct one to be selected due to its lower p-value. Please note the language chosen reflects the statistical inference for the network analysis; However, the mapping of inference to atmospheric behavior and known challenges around PM2.5 source apportionment is important and discussed.

• Line 149: Detrending a dataset does not always correct for non-stationarity. Please cite a reference for its appropriateness here.

Response: The following references are added to the manuscript:

(Papagiannopoulou et al., 2017, 2016)

• Line 157: What is a lag used for in this case?

Response: The optimal chosen lag is based on the Akaike Information Criterion (AIC). This ensures the model will be stable, and we found a value of 7 (unitless) was appropriate in our study.

• Lines 345-348: It is not clear in the text why this basic description of a directed graph is in the middle of the section on Granger Causality?

Response:

The previous undirected graph indicates the existence of correlation. The directed graph shows the direction of potential causal mechanism (e.g. pollution from A leads the pollution from B, possibly indicating a transport process). We use the Granger causality to build the directed graph and go further to analyse its stability using hierarchical trophic coherence.

Section 2.4 The language regarding trophic coherence and sources/sinks is unreferenced and thus assumed to be an innovation of this work. All cities in this region are known to be atmospheric sources of PM2.5 due to anthropogenic and natural processes. The discussion here and elsewhere in the manuscript of only some cities classified as sources of PM2.5 is at odds with reality and should be clarified further.

Response: Yes, you are right!! But we looked at only one year to demonstrate the usefulness of our approach in the first instance. About site selection, again to demonstrate the usefulness of our approach we decided to conduct a small study first. Besides, we were trying to show the impact of regional sources on PM2.5 level in the UK, therefore decided to focus more on urban background sites. In some cities such as London, Birmingham, and Chesterfield we have various urban background sites, since we were not interested in in inferring causality between sites across a city, the number of stations reduced in different cities. Focusing on larger networks and smaller regions is

something that can follow in future studies.

• While tropic coherence has been shown to provide interesting results for the analysis of food webs, it's not clear from the text that this is the most appropriate method for assessing important vertices in a causal graph for PM2.5. Why was it used here?

Response: Trophic coherence is used to analyse the general directionality coherence of the causal network. If we had perfect coherence (q=0), then there is a source of pollution that is affecting others. If we had perfect incoherence (q=1), then all the cities are polluting each other equally. This gives us an idea of both the nature and geography of the transport ecosystem for different seasons, as well as its stability.

• On line 39 in the abstract, the authors claim that winter is the most coherent of the seasons. They attribute this to meteorological features like wind speeds and inversions. Table 3 shows that summer has nearly the same incoherence factor as winter, and yet this is not discussed at all in the manuscript. Given the vastly different meteorological features in summertime, does this influence the interpretation that meteorology is a driving factor?

Response: (Updated in the Paper to the Results section): Table 3 shows a similar incoherence factor for winter and summer. With a q value of 0.3-0.4, this suggests having a single source of pollution and similar stability. The summer and winter periods have similar values but different sources. Figure 4B (summer) suggests that the source of network are Liverpool, London Road, Southampton, Norwich, and Birmingham. While, in winter (Figure 4 D), the sources of network are Chesterfield, Manchester, Preston, London Bexely. Winter is also inferred to be represented as a national network, while summer is more local.

We would also add that, according to the figures 2-4, the less incoherent network was witnessed during wintertime comparing to the rest of seasons. During springtime, due to well mixing of the lower atmospheric layer, the network was well formed. On the other hand, during wintertime, the meteorology is characterized by frequent inversions, form-

ing an efficient obstacle for the distribution and homogenization of PM. As a result, only tight spatially embedded parts of network (below 100Km with the highest percentage of restored network) could 'withstand' meteorological influences and larger distances across the network (above 100Km) started to collapse from a network perspective. In winter time, the plausible reason of connecting the cities out of the initial network (81% of connected cities were out of the initial network with distance above 200 Km) might be higher average seasonal wind speeds (in all studied stations), probably due to the balance among greater dilution and shorter transport times at higher wind speeds, which allows less time for PM dispersion and deposition over further distances(Harrison et al., 2012).

Discussion In general, this discussion seems incomplete and largely conjecture without appropriate referencing or analysis presented herein.

âǍć Lines 378-381: What in the previous analysis indicates that meteorological conditions and diurnal emissions from regional sources dominate? I acknowledge that this is known the be case throughout the field a priori, but it's not clear how this analysis in this manuscript leads to that conclusion.

Response: Harrison et al., (2012) showed how meteorological parameters with the focus on wind speed and wind direction can influence PM2.5 level in the UK and provide insight into the origin the measured PM2.5 concentrations. Based on their study, a notable consistency in the patterns across the UK exists. When the winds are coming the south-southeast clockwise through to north, the PM2.5 concentrations are generally lower than the annual average value, while when the winds are coming from the northeast through to southeast, the PM2.5 concentrations are higher than the annual average value (Harrison et al., 2012). In addition, secondary aerosols secondary sources are dependent on meteorological conditions, and the abundance of precursors, that can have a significant contribution in PM2.5 concentrations in the UK, where a large proportion of transboundary secondary PM2.5 are transferred from different parts of Europe (AQEG, 2012; Vieno, et al., 2016). As a result, one plausible reason

of connection within a network can be common transboundary sources.

• Lines 399-409: How are these relationships known without detailed trajectory modelling? The fact that the causal networks are consistent with known transport mechanisms is interesting but does not provide evidence for these sweeping assessments of pollutant transport.

Response: The reviewer is correct. We did not conduct detailed trajectory modelling but tried to interpret our casual network based on previous studies and try to explain the reason behind generated clusters in south and north of the UK. Inferring causal mechanisms from data is not new, and the fact that our findings corroborate with previous detailed modeling and known qualitative causal mechanisms we hope demonstrates the usefulness of this approach.

• Lines 410-415: There is not evidence presented in this manuscript that this transport mechanism is attributable to PM2.5 variability in the Northern UK.

Response: Our results (Figures 2-4) showed coherence throughout the patterns across Group A and Group B in the UK. Based on previous studies, high PM2.5 concentrations in Group B (southern sites) are more attributed to winds from the east through to southeast, which are often attributed to a blocking high pressure over the Nordic countries, giving rise to a south-easterly or easterly air flow that cause transportation of emissions from eastern Europe, northern Germany, and the Belgium and Netherlands to the southern cities in the UK. High PM2.5 concentrations in Group A (northern cities or close to northern part) are attributed to the winds blowing from the northeast through to east, drawing air flow (likely to start blowing when a low pressure runs up the English Channel) northward across European emission sources (to mainly be emission sources of precursors of secondary PM), out into the North Sea, then reaching northern parts of the UK from a north-easterly direction (Barry, R.G., Chorley, R.J., 2010; Harrison et al., 2012).

Conclusions • Line 423-425: The authors haven't shown any results related to predicting future PM2.5 between cities. Response: We apologise for this confusion and have removed the word "predict". Past values have information which is statistically significant to future values. We use this in our causal analysis, but we do not make active predictions, only statistical inferences.

• Line 439-431: This work is not the first to demonstrate that meteorology drives much of the variability in PM2.5. It is not clear from this manuscript how connections beyond conjecture can be made to meteorological variability and PM2.5. Response: This is a fantastic point and as we touched on in our earlier response, our hope is to link our network findings back to meteorology findings and models. What we are keen to show and add to is that there is a topological aspect, which highlights the complex web of cascade pollution transport between cities.

Minor Comments • Line 78: "Atmospheric particulate matters" should be "Atmospheric particulate matter" Response: Corrected and highlighted in text.

• Figure captions should contain more detail regarding the content of the figured. For example in Figure 2, what do edge thicknesses correspond to? Response: In the revised version of figures, we will try to add more details. • Line 232: "%50" should be "50%". Response: Corrected and highlighted in text.

• Line 393: "Primary" should be "Primary". Response: Corrected and highlighted in text.

References: AQEG, 2012. Fine Particulate Matter (PM2.5) in the UK. AQEG, 2005. Particulate Matter in the United Kingdom. London. Barry, R.G., Chorley, R.J., 2010. Atmosphere, Weather and Climate, ninth ed. ed. Routledge, Abingdon. DEFRA, 2015. Department for Environment, Food and Rural Affairs, United Kingdom. Gehrig, R., Buchmann, B., 2003. Characterising seasonal variations and spatial distribution of ambient PM10 and PM2.5 concentrations based on long-term Swiss monitoring data. Atmos. Environ. 37, 2571–2580. https://doi.org/10.1016/S1352-2310(03)00221-8 Hammer, O., Harper, D., Ryan, P., 2001. PAST: Paleontological Statistics Software

Package for Education and Data Analysis. Palaeontol. Electron. 4, 1–9. Harrison, R.M., Laxen, D., Moorcroft, S., Laxen, K., 2012. Processes affecting concentrations of fine particulate matter (PM2.5) in the UK atmosphere. Atmos. Environ. 46, 115–124. https://doi.org/10.1016/J.ATMOSENV.2011.10.028 Munir, S., 2016. Analysing temporal trends in the ratios of PM2.5/PM10 in the UK. Aerosol Air Qual. Res. 17, 34–48. https://doi.org/10.4209/aaqr.2016.02.0081 Papagiannopoulou, C., Decubber, S., Miralles, D.G., Demuzere, M., Verhoest, N.E.C., Waegeman, W., 2017. Analyzing Granger Causality in Climate Data with Time Series Classification Methods BT - Machine Learning and Knowledge Discovery in Databases, in: Altun, Y., Das, K., Mielikäinen, T., Malerba, D., Stefanowski, J., Read, J., Žitnik, M., Ceci, M., Džeroski, S. (Eds.), . Springer International Publishing, Cham, pp. 15–26. Papagiannopoulou, C., Miralles, D., Verhoest, N., Dorigo, W., Waegeman, W., 2016. A non-linear Granger causality framework to investigate climate–vegetation dynamics. Geosci. Model Dev. Discuss. 1–24. https://doi.org/10.5194/gmd-2016-266 Software, Q.M., 2019. Eviews, Version 11.

---

## Author Comment (AC2) · 29 May 2020

29/05/2020 Dear GMD editorial board, Subject: Submission of revised paper gmd-2019-342 Thank you for your email dated 1st May 2020 enclosing the reviewers' comments. We have carefully reviewed the comments and have revised the manuscript accordingly. Our responses are given in a point-by-point manner below. Changes to the manuscript are highlighted. We hope the revised version is now suitable for publication and look forward to hearing from you in due course. Sincerely, Jong R. Kim, PhD, PE, CMP Professor Department of Civil and Environmental Engineering School of Engineering and Digital Sciences Nazarbayev University Nursultan (Astana), Kaza-

khstan

Phone: +7 7172 70 9136 Mobile: +7 775 069 1735

gmd-2019-342-RC2#

• the small subset of data used (e.g. 15 sites in 14 cities) Response: Yes, you are right!! But to demonstrate the usefulness of our approach we decided to conduct a small study first. Besides, we were trying to show the impact of regional sources on PM2.5 level in the UK, therefore decided to focus more on urban background sites. In some cities such as London, Birmingham, and Chesterfield we have various urban background sites, since we were not interested in in inferring causality between sites across a city, the number of stations reduced in different cities. Focusing on larger networks and smaller regions is something that can follow in future studies.

• the small temporal period considered (just 1 year) Response: Yes, you are right!! But we looked at only one year to demonstrate the usefulness of our approach in the first instance. • small about of parameters considered (solely PM observations and a national inventory of emissions) Response: As we mentioned in Conclusion, our study has some limitations. As a result, to have a better understanding of the network, evaluating a predictive network based PM2.5 model using meteorological parameters, and contributions from identified clusters in the UK, would be helpful and will be investigated in our future research.

• the lack of consideration of known variables that influence PM (e.g. meteorology) Response: As we mentioned in Conclusion, our study has some limitations. As a result, to have a better understanding of network, evaluating a predictive network based PM2.5 model using meteorological parameters, and contributions from identified clusters in the UK, would be helpful and will be investigated in our future research.

• The insufficient of placing this work in the context of existing related publications Response: We thank the reviewer for identifying further references to add to this work.

Based on the above issues raise, we change our paper as demonstrated in the following text.

(Updated in the paper to the introduction section): A number of studies have deployed a range of techniques to overcome challenges in computational performance of chemical transport models. Solving atmospheric chemical kinetics is a stiff numerical problem, with choice of solvers used reflecting the need to ensure numerical stability(Sandu and Sander, 2006). Consequently, the integration of the chemical kinetics takes 50%–90% of the computational cost of an atmospheric chemistry model such as GEOS-Chem (Eastham et al., 2018; Hu et al., 2018; Nielsen et al., 2017). Dynamical reduction (adaptive solvers) in solving the chemical mechanism was previously demonstrated to increase the efficiency of the integration at the expense of a reduction in accuracy (Cariolle et al., 2017). Other attempts to reduce the computationally of chemical kinetics include repro-modelling (approximation of the chemical kinetics using polynomial functions)(Turányi, 1994), quasi-steady state approximation(Whitehouse et al., 2004), and separation of fast and slow species(Young and Boris, 1977). Other studies use reduced chemical mechanisms with fewer species (Kelp et al., 2018; Whitehouse et al., 2004). Recent attempts have also used machine learning to replace the use of traditional integrators (Porumbel et al., 2014). For example, using a neural network emulator for an atmospheric chemistry box model, an order-of-magnitude speed up was found, but the new implementation suffered from rapid error propagation when applied over multiple time steps (Kelp et al., 2018). Numerical emulators have also been used to directly forecast air pollution concentrations across future time steps (Mallet et al., 2009). This approach was also applied in chemistry– climate simulations with the focus on model forecasting of time averaged concentrations of selected species such as OH (hydroxyl radical), and O3 (ozone) over timescales of days to months(Nicely et al., 2017; Nowack et al., 2018). Keller and Evans (2019) studied the replacement of suitably trained machine-learning based approach (random forest regression) for the gas-phase chemistry in atmospheric chemistry transport models (GEOS-Chem). As noted within this particularly study, this approach suffers also from some limitations including; (a) being only applied within the range of data used for the training, (b) studying scenarios with significant changes in the emissions (being outside of used data for the training) can lead to inaccurate predictions by the model, and (c) machine learning algorithm may not capture model resolution caused by the non-linear nature of chemistry (the numerical solution of chemical kinetics is resolution-dependent)(Keller and Evans, 2019).

• Many years of data are freely available for hundreds of sites across the UK. Why have only 15 sites from 14 locations been considered here? Why were these 15 sites (listed in the SI) chosen above the hundreds of others? Are the authors saying they are representative of the sites? Why only 15 sites? Is it due to limitations in a computer resource or does this approach not scale well? If this is the case could a representative average be taken of the sites in a city? Why has a mixture of types been used (e.g. Traffic urban, Background suburban, Background Urban, and Industrial Urban)? What impact could the choice of data set have on the conclusions? Response: Yes, you are right!! But we looked at only one year to demonstrate the usefulness of our approach in the first instance. About site selection, again to demonstrate the usefulness of our approach we decided to conduct a small study first. Besides, we were trying to show the impact of regional sources on PM2.5 level in the UK, therefore decided to focus more on urban background sites. In some cities such as London, Birmingham, and Chesterfield we have various urban background sites, since we were not interested in in inferring causality between sites across a city, the number of stations reduced in different cities. Focusing on larger networks and smaller regions is something that can follow in future studies.

• All of these issues need to be explored through sensitivity testing. Response: Our data enables us to construct an undirected correlation and a directed Granger causality network, using PM2.5 concentrations in 14 UK cities over a year-long period. We show for both reduced-order cases, the UK is divided into two northern and southern connected city communities, with greater spatial embedding in spring and summer. We
go on to infer stability to disturbances via the network trophic coherence parameter, whereby we found that winter had the greatest vulnerability. As a result of our novel graph-based reduced modelling, we are able to represent high-dimensional knowledge into a causal inference and stability framework. Our statistical p values demonstrate confidence in results which embeds robustness. We would like to expand this further, but we cannot do it at the end of an 8-month manuscript review due to realistic researcher employment and practical reasons. Had this been raised much earlier, we may have had the resources.

• This analysis must be expanded in breadth (# sites) and depth (e.g. years of data) if it is to be considered a case study or proof of concept is. Response: Yes, you are right!! But we looked at only one year to demonstrate the usefulness of our approach in the first instance. About site selection, again to demonstrate the usefulness of our approach we decided to conduct a small study first. Besides, we were trying to show the impact of regional sources on PM2.5 level in the UK, therefore decided to focus more on urban background sites. In some cities such as London, Birmingham, and Chesterfield we have various urban background sites, since we were not interested in in inferring causality between sites across a city, the number of stations reduced in different cities. Focusing on larger networks and smaller regions is something that can follow in future studies.

• How many air quality events happened in this year? What does this approach show about these events? Often pollution events in the UK are synoptic in scale, so would affect multiple cities. It would be interesting to see what information could be drawn out from this analysis. Response: The paper is showing correlations and potential causal pathways inferred from data across UK cities. We show for both reduced-order cases, the UK is divided into two northern and southern connected city communities, with greater spatial embedding in spring and summer. We have not studied certain potential air quality events and are they synoptic in scale and how they can influence different cities. In further investigations it would be an interesting lead to follow.

• The discussion needs to be expanded to explain what the pros and cons of this approach are over the generally taken approaches (e.g. chemical transport models). Also consider this approach against other non-explicit approaches taken in this field (e.g. Nowack et al 2020, Keller et al 2019). ML methods that consider much more input data (e.g. meteorology, emissions, chemistry) have already been demonstrated in a much more thorough way within this journal (e.g. Keller et al 2019). Also make it clear what new information that this approach will provide and why this would be advantageous above these other approaches or complementary to them.

Response: (Updated in Paper to the discussion section): The general framing of our approach is at the national level, trying to demonstrate (via a data-driven correlation and causal network), the statistical relationship between data from multiple cities. This data-driven, low-dimensional, network enables us to examine seasonal trends and infer root causal mechanisms. We believe this approach requires evaluations across multiple scales. Nonetheless, we believe this approach will offer an additional approach to traditional models where inference of causality remains challenging. Of course, what our model lacks is the relationship back to the physical flow models, and our future work will incorporate this. Machine Learning models are used to predict, but we are here to infer causality and demonstrate topological patterns via the network. We also respond in more detail (not in paper): As the reviewers are well aware, chemical transport models require emission inventory data (local or regionally originated) and a meteorological core to predict the dispersion and deposition of pollutants such as PM2.5. Beside the notable amount of required data, high performance computing [HPC] platforms are required to deploy and evaluate model outputs, not least including experience with the pre and post processing software environments. In the current study we attempt to investigate the behaviour of PM2.5, using a 2-dimensional (2D) network constructed from observational data alone. This leads to, first, the correlation network, and then a causation network. We can identify two things of note. (1) is the presence of root causations of pollution in certain seasons across a large region of UK, and (2) the stability of the transport network to potential disturbances. Both

provide a level of simplistic insight at a very low complexity. (Updated in the paper to the introduction section): Previous studies tried different techniques to overcome difficulties in simulation of atmospheric chemistry transport processes. One of the faced challenges is about computationally expensive nature of these models. The numerical solution of chemical kinetics is computationally expensive due to the numerically stiff equations needs implicit integration schemes (like Rosenbrock solvers) to ensure numerical stability(Sandu and Sander, 2006). Consequently, the integration of the chemical kinetics takes 50%–90% of the computational cost of an atmospheric chemistry model such as GEOS-Chem (Eastham et al., 2018; Hu et al., 2018; Nielsen et al., 2017). Involving dynamical reduction (adaptive solvers) in the chemical mechanism was previously tried methods to increase the efficiency of the integration, associated with accuracy reduction(Cariolle et al., 2017). Some other previous attempts to reduce computationally cost chemical kinetics are repro-modelling (approximation of the chemical kinetics using polynomial functions)(Turányi, 1994), quasi-steady state approximation(Whitehouse et al., 2004), and separation of fast and slow species(Young and Boris, 1977). Some other studies tried to simplify the chemistry causing a decrease in the number of species and reactants (Kelp et al., 2018; Whitehouse et al., 2004). Using machine learning, the chemical integrator was replaced for other chemical systems and were faster than solving the ODEs (chemical systems like those found in combustion) (Porumbel et al., 2014). Recently, using a neural network emulator for an atmospheric chemistry box model, an order-of-magnitude speed ups was found, but it suffered from rapid error propagation when applied over multiple time steps (Kelp et al., 2018). Machine learning emulators have also been tried to directly forecast the air pollution levels in future time steps(Mallet et al., 2009). This approach was also applied in chemistry– climate simulations with the focus on model forecasting of time averaged concentrations of selected species such as OH (hydroxyl radical), and O3 (ozone) over timescales of days to months(Nicely et al., 2017; Nowack et al., 2018). Keller and Evans (2019) studied the replacement of suitably trained machine-learning based approach (random forest regression) for the gas-phase chemistry in atmospheric

chemistry transport models (GEOS-Chem) compares well to the standard modelling. The new approach was to forecast the concentration of each transported specie including NOx and O3 (Keller and Evans, 2019), made it comparable to previous attempts in speeding up the solution of the chemical kinetics through more efficient integration. Current approach suffers also from some limitations including; (a) being only applied within the range of used data for the training, and applying the method outside of this range can cause inaccurate outputs, (b) studying scenarios with significant changes in the emissions (being outside of used data for the training) can lead to inaccurate predictions by model, and (c) machine learning algorithm may not capture model resolution caused by the non-linear nature of chemistry (the numerical solution of chemical kinetics is resolution-dependent)(Keller and Evans, 2019).

Line 102 - This is sentence is far too strong. The authors have not demonstrated that this technique could be this useful. "This is a timely study as strategic investments in national and local air quality monitoring networks require an evaluation on the usefulness, or not, of network design" Response: We are not sure why the reviewer feels this sentence is too long or is invalid. We are not claiming the method can be used immediately but have submitted this study for scientific peer review. We use a case study of selected sites to confirm whether inferred causality makes sense based on known relationships in PM2.5 and variable conditions and transport mechanisms. Much like studies demonstrating the potential use of ML variants of existing models are far from wide scale adaption, for reasons discussed in the literature, we feel our study demonstrates potential for the method presented here.

 ć Line 112 - See earlier point about the limitations of data used. Response: Yes, you are right!! But we looked at only one year to demonstrate the usefulness of our approach in the first instance. About site selection, again to demonstrate the usefulness of our approach we decided to conduct a small study first. Besides, we were trying to show the impact of regional sources on PM2.5 level in the UK, therefore decided to focus more on urban background sites. In some cities such as London, Birmingham,

and Chesterfield we have various urban background sites, since we were not interested in in inferring causality between sites across a city, the number of stations reduced in different cities. Focusing on larger networks and smaller regions is something that can follow in future studies.

• Line 117 - Was this emissions inventory used for the same year? Given more details here. Response: At the current stage we decided not to use NAEI data and will be deleted from revised version in the materials & methods section.

• Line 163 - What is the link to the broader picture here? If we accept that a background urban site in Manchester can be used to predict future concentrations in Preston, is the suggestion that something like this could be done for other sites and a national level of prediction gained at a computational cheap cost? Response: Yes, there is a possibility that we can understand how UK cities cross-pollute across regional and national distances, and we are now investigating it in our ongoing research.

• Line 424 - There is no supporting evidence given for this predictive capacity. Response:

We apologise for this confusion and have removed the word "predict". Past values have information which is statistically significant to future values. We use this in our causal analysis, but we do not make active predictions, only statistical inferences.

Technical points

• Title / other text - Some lines are highlighted blue. Why? Response: We had a minor revision based on editor's comment asked us to revise and highlight revised parts, which is reason for blue highlighted parts through the manuscript. • Table 2 - typo in units? (Kelvin metres instead of kilometres?) Response: Corrected in the text. • Figure 1 - resolution needs to be improved. Most of UK coastline should be shown for context. Which map layer was used? Response: The resolution will be improved in revised manuscript. • Figure 2 - resolution needs to be improved. Response: The

resolution will be improved in revised manuscript.

• Figure 3 - resolution needs to be substantially improved. Axis labels are not large enough or readable. Response: The resolution will be improved in revised manuscript.

• Line 9 - Specify the region over which the deaths occur. Europe? Global? Response: Global • Line 32 - Have people not already been successful in doing this? (e.g. the climate science community - Nowack et al 2020) Response: The causal networks in Nowack et al are causal networks between confounding factors (an analogy here would be: causal network between industrialisation, production, and air pollution). Our causal networks are geographic, in that we are looking at whether cities causally influence each other. • Line 34 - "12 month", "52 week" are used interchangeably in the manuscript: would a single phrasing (e.g. year-long) be easier for the reader? Response: Sure!! Corrected and highlighted through the text. • Line 35 - Sentence does not scan. superfluous "two" and "a"? Response: Checked and corrected in the text.

• Line 80 - Globally? The focus of the article is the UK, so you need to be more specific. Response: The following part will be added to the text;

(Updated in the paper to the introduction): In the UK, long-term exposure to PM from anthropogenic sources has an impact on the equivalent of around 29,000 deaths a year(COMEAP, 2010; Gowers etal., 2014). Also, short-term exposure to air pollution events can increase the daily emergency hospital admissions (for cardiovascular and respiratory conditions) and mortality(Macintyre et al., 2016). Focusing on two air pollution events (12– 14 March and 28 March–3 April 2014) with the highest PM2.5 concentrations, about 600 deaths were brought forward from short-term PM2.5 exposure, representing 3.9% of total all-cause death during these 10 days.

• Line 112 - This time-split would be better placed in a table and in the SI. It is not a very readable way of presenting this information. Response: Table S2 including time-split data created and added to the supplementary material.

• Line 116 - Was data downloaded for the same period. This sentence does not need to start with" also". Response: At the current stage we decided not to use NAEI data, it was left here by accident and will be deleted from revised version. • Line 393 - formatting error of "y" in word primary. Response: Corrected and highlighted in text. • Data availability - Add a data availability section. Include specific details on which version of data was used and how to access this. Response: Data availability section will be added to the revised version, including all above-mentioned details. (Updated in the Paper to the data availability section): Data availability section: The measurement were hourly based taken from UK Automatic Urban and Rural Network (AURN) (https://uk-air.defra.gov.uk/data/openair). More information about UK Automatic Urban and Rural Network is available online from the DEFRA website(DEFRA, 2015). Data coverage were checked to have minimum missing data and having at least 75% of hourly based measured data for all stations, before averaging the hourly PM2.5 concentration. Only available data for 20 hours a day were averaged. While zero, NAN, and negative values were removed from the data set, and if the remained values were at least 20 hours a day, we averaged it representing the daily PM2.5 concentration, if not we report that day as NAN. For PM2.5 measurement in UK monitoring system, for daily and hourly averaged concentrations, the instrument of FAI SWAM 5a was used be Defra (Defra approved instrument) which was certified to MCERTS (The Environment Agency's Monitoring Certification Scheme) for UK particulate Matter, and also certified to MCERTS for CAMs (Continuous Ambient Measurement Systems) of particulate Matter (https://uk-air.defra.gov.uk/networks/monitoring-methods?view=mcerts-scheme) (DEFRA, 2015). Reference equivalent method FDMS (Filter Dynamic Measurement System) is used for PM2.5 measuring at studied stations, which is allowed by EU for regulatory purposes(AQEG, 2012).

Characterisation of PM2.5 temporal variability is important when it can help us to observe the high levels of pollutant causing health problems. Due to the data unavailability in the UK, it is not possible to conduct the historically long-term temporal trend analysis of PM2.5 (AQEG, 2005). Based on the AQEG (2012) report, there are no

monitoring stations with long term (> 5years) using reference equivalent instruments for PM2.5 monitoring. From 2008–2009 onward, with the increase in the number of monitoring stations using reference equivalent method (such as FDMS allowed by EU for regulatory purposes) it is possible to study the temporal changes in PM (PM10 & PM2.5) in the UK (Munir, 2016). Minimum performance requirement for PM10 & PM2.5 analysers were outlined in standard method of EN12341:2014 PM10 and PM2.5 (EN16450:2017 Automatic PM analysers). These methods are proposed to ensure that measurement methods are complying with the DQO (Data Quality Objectives) set down in the Ambient Air Quality Directive (2008/50/EC) and in the amending Directive (EU) 2015/1480. The monitoring techniques used the UK's AURN for PM10 & PM2.5 ( with the exception of the automatic PM10 analysers) are; Tapered Element Oscillating Microbalance, Beta Attenuation monitor, Gravimetric monitor, Filter Dynamics Measurement System (FDMS), Optical light scattering, and Fine dust Analysis System (FIDAS)(DEFRA, 2015). • References - remove full stops at starts of lines. Response: Checked and corrected in the text.

• S1 list - please update the information to be in a table format. The current format is unwieldy, not read-friendly. Response: Table S including the studied Monitoring stations' data created and added to the supplementary material.

• Citations Keller, C. A. and Evans, M. J.: Application of random forest regression to the calculation of gas-phase chemistry within the GEOS-Chem chemistry model v10, Geosci. Model Dev., 12, 1209–1225, https://doi.org/10.5194/gmd-12-1209-2019, 2019. Nowack, P., Runge, J., Eyring, V. and Haigh, J.D., 2020. Causal networks for climate model evaluation and constrained projections. Nature communications, 11(1), pp.1-11. Interactive comment on Geosci. Model Dev. Discuss., https://doi.org/10.5194/gmd-2019-342, 2020. Response: Proposed references will be added to the reference section and will be addressed in our manuscript in revised version.

References: AQEG, 2012. Fine Particulate Matter (PM2.5) in the UK. AQEG, 2005.

[Figure]

Particulate Matter in the United Kingdom. London. Barry, R.G., Chorley, R.J., 2010. Atmosphere, Weather and Climate, ninth ed. ed. Routledge, Abingdon. COMEAP, 2010. The Mortality Effects of Long-term Exposure to Particulate Air Pollution in the United Kingdom. DEFRA, 2015. Department for Environment, Food and Rural Affairs, United Kingdom. Gehrig, R., Buchmann, B., 2003. Characterising seasonal variations and spatial distribution of ambient PM10 and PM2.5 concentrations based on long-term Swiss monitoring data. Atmos. Environ. 37, 2571–2580. https://doi.org/10.1016/S1352-2310(03)00221-8 Gowers, A.M., Miller, B.G., Stedman, J.R., 2014. Estimating Local Mortality Burdens Associated With Particulate Air Pollution [©] Crown copyright 2014, licenced under the Open Government Licence (OGL). Hammer, O., Harper, D., Ryan, P., 2001. PAST: Paleontological Statistics Software Package for Education and Data Analysis. Palaeontol. Electron. 4, 1–9. Harrison, R.M., Laxen, D., Moorcroft, S., Laxen, K., 2012. Processes affecting concentrations of fine particulate matter (PM2.5) in the UK atmosphere. Atmos. Environ. 46, 115–124. https://doi.org/10.1016/J.ATMOSENV.2011.10.028 Macintyre, H.L., Heaviside, C., Neal, L.S., Agnew, P., Thornes, J., Vardoulakis, S., 2016. Mortality and emergency hospitalizations associated with atmospheric particulate matter episodes across the UK in spring 2014. Environ. Int. 97, 108–116. https://doi.org/https://doi.org/10.1016/j.envint.2016.07.018 Munir, S., 2016. Analysing temporal trends in the ratios of PM2.5/PM10 in the UK. Aerosol Air Qual. Res. 17, 34–48. https://doi.org/10.4209/aaqr.2016.02.0081 Papagiannopoulou, C., Decubber, S., Miralles, D.G., Demuzere, M., Verhoest, N.E.C., Waegeman, W., 2017. Analyzing Granger Causality in Climate Data with Time Series Classification Methods BT - Machine Learning and Knowledge Discovery in Databases, in: Altun, Y., Das, K., Mielikäinen, T., Malerba, D., Stefanowski, J., Read, J., Žitnik, M., Ceci, M., Džeroski, S. (Eds.), . Springer International Publishing, Cham, pp. 15–26. Papagiannopoulou, C., Miralles, D., Verhoest, N., Dorigo, W., Waegeman, W., 2016. A non-linear Granger causality framework to investigate climate–vegetation dynamics. Geosci. Model Dev. Discuss. 1–24. https://doi.org/10.5194/gmd-2016-266 Software, Q.M., 2019. Eviews,

Version 11.